# Autonomous mesoscale positioning emerging from myelin filament self-organization and Marangoni flows

Arno van der Weijden [1,2], Mitch Winkens[1,2], Sandra M. C. Schoenmakers[1], Wilhelm T. S. Huck [1] & Peter A. Korevaar [1✉]

Out-of-equilibrium molecular systems hold great promise as dynamic, reconfigurable matter that executes complex tasks autonomously. However, translating molecular scale dynamics into spatiotemporally controlled phenomena emerging at mesoscopic scale remains a challenge—especially if one aims at a design where the system itself maintains gradients that are required to establish spatial differentiation. Here, we demonstrate how surface tension gradients, facilitated by a linear amphiphile molecule, generate Marangoni flows that coordinate the positioning of amphiphile source and drain droplets floating at air-water interfaces. Importantly, at the same time, this amphiphile leads, via buckling instabilities in lamellar systems of said amphiphile, to the assembly of millimeter long filaments that grow from the source droplets and get absorbed at the drain droplets. Thereby, the Marangoni flows and filament organization together sustain the autonomous positioning of interconnected droplet-filament networks at the mesoscale. Our concepts provide potential for the development of non-equilibrium matter with spatiotemporal programmability.

[1] Institute for Molecules and Materials, Radboud University, Heyendaalseweg 135, Nijmegen 6525 AJ, The Netherlands. [2] These authors contributed equally: Arno van der Weijden, Mitch Winkens. ✉email: p.korevaar@science.ru.nl

L iving matter is driven by well-regulated systems that coordinate self-assembly, process information, and transmit signals to organize components in time and space[1]. Functions that emerge from these systems offer great inspiration for the design of self-organizing systems that display behavior, such as motion[2–5], pattern formation[6], or complex shape transformations[7–10]. Whereas the molecular assembly of static, equilibrium structures is well understood, the development of functional matter with life-like features requires a variety of out-of-equilibrium mechanisms that enable programmable self-organization of the components. To this end, temporal programming can be established by coupling of the molecular assembly to dissipative chemical reactions[11–18]. For example, in the construction of the cellular cytoskeleton, microtubules only appear within a timeframe that integrates tubulin activation and (de)polymerization rates[19]. This concept has inspired the temporal programming of synthetic structures, varying from nanofibers[20–22] to nanoparticle aggregates[23], micelles[24], microdroplets[25], and macroscale gels[26] that form and fall apart as dictated by the kinetics of the underlying chemical reactions.

Spatial programmability in self-organizing systems, however, is more challenging to establish, especially if one aims to translate interactions at the (supra)molecular level into spatiotemporally controlled phenomena at mesoscopic length scales. Typically, spatial control in synthetic systems involves external stimuli that contain spatial information, such as concentration gradients[27], external fields[28–30], or localized photocontrol[31], although self-organizing patterns emerging from initially homogeneous conditions have been established via reaction diffusion[32,33] and convection[34]. Furthermore, chemical gradients—either applied top-down or emerging spontaneously—can drive motion in self-organization processes[35–38]. For example, Marangoni flows driven by surface tension gradients caused by dissolution of surface-active chemicals have been reported to direct the ordering of swarms of floating droplets or camphor boats at aqueous interfaces[39,40]. Alternatively, the self-organization of active micro-droplets that either repel or attract each other can be directed by dissolution of the droplet content in micellar aqueous solutions[41–43]. Together, the variety of spatiotemporal dynamic behavior that has been established in these systems demonstrates that fluxes that maintain concentration gradients serve as a means to coordinate spatial differentiation—prerequisite to obtain spatial self-organization. We reasoned that fluxes that coordinate the positioning of their own driving forces enable a design principle to program positioning routines in self-organizing systems.

We introduce a system that exemplifies the concept of positioning through self-sustained gradients via Marangoni flows: drain droplets deplete amphiphile molecules from an air–water interface and thereby sustain a Marangoni flow that repels the drain droplet from the amphiphile source. At the same time, the Marangoni flow drags self-assembled amphiphile filaments towards these drains, and in turn, absorption of these filaments at the drain generates attractive forces that maintain the positioning of the drain droplets. First, we demonstrate how millimeter-long filaments emerge from an amphiphile source droplet that floats on water, as they originate from a buckling instability of amphiphile bilayers in the droplet and get extruded by the Marangoni flows. Second, we show experimentally and rationalize with a model how the positioning of source and drain droplets relies on the balance between the amphiphile release and depletion kinetics, and the filament absorption. Finally, we demonstrate how a well-balanced coupling between the Marangoni flows and the filament absorption, which generate repulsive and attractive forces, respectively, drives the autonomous positioning of interconnected droplet networks.

## Results

**Buckling of amphiphile bilayers generates myelin filaments.** The design of our system relies on the amphiphile tetra(ethylene glycol) monododecyl ether ($C_{12}E_4OH$), which forms long filament structures and, at the same time, shows interesting surfactant release/depletion dynamics at air–water interfaces. Linear amphiphiles such as $C_{12}E_4OH$ are known to form a laminar phase of closely packed bilayers in concentrated mixtures with water[44]. At the edge of an amphiphile droplet, the spaces in between these bilayers take up more water; the resulting pressure forces the packed bilayers to buckle and filaments, consisting of packed cylinders of amphiphile bilayers, grow at the boundary of the amphiphile droplet (Fig. 1a)[45]—a phenomenon that has been reported for other linear amphiphiles[46–50] and block-copolymers[51,52] as well. Small-angle X-ray scattering studies on $C_{12}E_4OH$-based structures have shown that these protrusions—also called "myelins"—consist of packed cylinders of amphiphile bilayers, separated by water with a spacing in the order of 10 nm[45]. To initiate the assembly of $C_{12}E_4OH$ into filaments, a 0.5 μL source droplet of 40 v/v% $C_{12}E_4OH$ in a water/ethanol mixture was deposited on the air–water interface of an aqueous sodium alginate solution (see "Methods"). Optical microscopy images reveal a blebbed surface of the droplet and after a few minutes multiple, ~10–50-μm-thick filaments emerge, that grow in all directions and extend for several millimeters (Fig. 1b, c and Supplementary Movie 1). As the amphiphile has a density <1 g mL$^{-1}$, the droplet as well as the filaments float at the air–water interface.

**Marangoni flows extrude filaments from source droplet.** Upon deposition of the amphiphile source droplet, the surface tension drops instantly from 72 mN m$^{-1}$ (pure water) to 29 mN m$^{-1}$ (Supplementary Fig. 1), indicating that individual amphiphile molecules are released from the droplet. The surface concentration of these amphiphiles approaches saturation, as a surface tension of 27 mN m$^{-1}$ has been reported in the literature for $C_{12}E_4OH$ beyond its critical micelle concentration[53]. The initial decrease in surface tension reversed over a couple of hours, as amphiphiles slowly dissolved into the underlying aqueous phase (Supplementary Fig. 1). The vacant sites at the air–water interface are supplied with new amphiphiles released from the source, generating a Marangoni flow that extrudes the filaments at a much faster rate (~13 μm s$^{-1}$ in Fig. 1c) compared to growth rates in the order of 0.1 μm s$^{-1}$ reported for typical myelin structures in the literature[50]. Furthermore, we note that a larger air–water interfacial area enhances the overall depletion rate of the amphiphile towards the aqueous phase, and thereby the extrusion rate of the filaments (Supplementary Fig. 1). To suppress the depletion kinetics towards the aqueous phase, and enhance the stability of the myelin filaments that progress over the air–water interface, sodium alginate was included in the aqueous phase (Supplementary Fig. 2).

The depletion of amphiphiles towards the underlying aqueous phase inspired us to introduce a drain droplet of a hydrophobic liquid that would locally deplete the amphiphiles from the interface, thus setting up a permanent gradient in amphiphile distribution at the air–water interface. Hence, a sustained Marangoni flow would be established from the amphiphile source droplet towards the drain droplet. When the rate of amphiphile release from the source is larger than depletion at the drain, the Marangoni flow pushes the source and drain droplets apart (Fig. 1d). At the same time, the filaments are extruded from the droplet by the Marangoni flow towards the drain, and get absorbed by the drain droplet upon arrival (Fig. 1e). As can be seen in Fig. 1f (Supplementary Movie 2), filament absorption

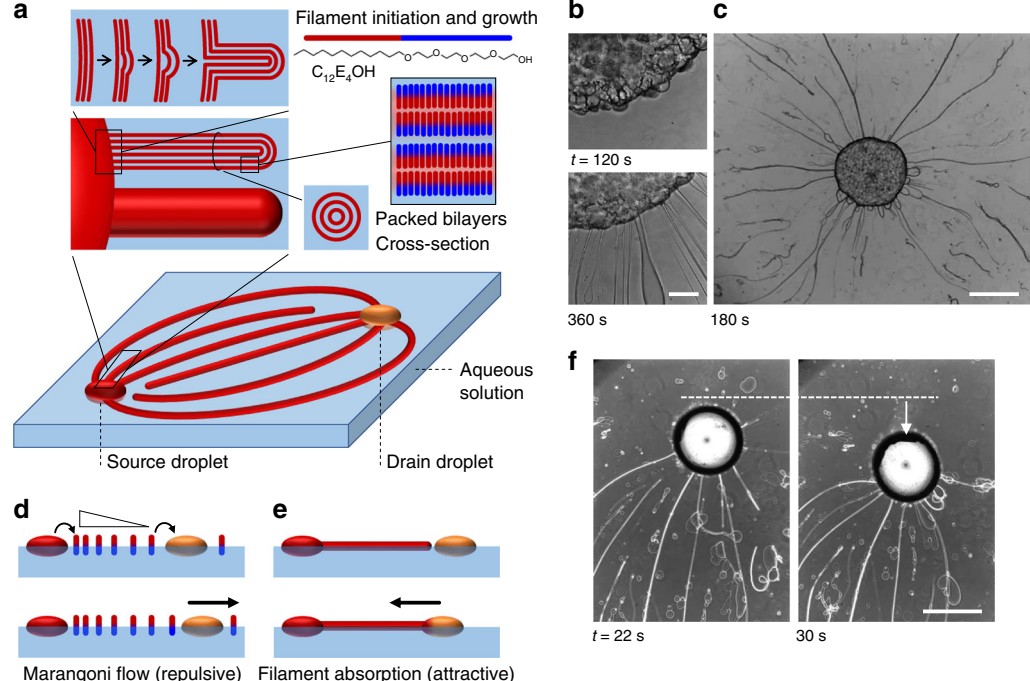

**Fig. 1 Marangoni flows and filaments direct positioning. a** In the amphiphile source droplet (red sphere), the $C_{12}E_4OH$ amphiphile forms closely packed bilayers. At the boundary of the droplet that is floating on water, the amphiphiles take up water (light blue), the packed bilayers buckle, and form long filaments consisting of bilayers. **b**, **c** Optical microscopy images of filament extrusion from the amphiphile source droplet, after deposition at an aqueous sodium alginate solution at $t = 0$ s. The scale bar in **b** represents 200 μm; the scale bar in **c** is 1 mm. **d** Concomitant to the filament growth, individual amphiphile molecules are released from the source droplet (red sphere) to the air–water interface. Subsequently, these amphiphiles are depleted at the drain droplet (orange sphere), such that a Marangoni flow emerges that pushes the source and drain droplets apart. **e** The drain, however, is pulled back to the source upon absorption of the filaments. A dynamic organization emerges when the repulsive (Marangoni flow) and attractive forces (filament absorption) match. **f** Upon filament absorption, the drain droplet (deposited at $t = 0$ s) is pulled towards the source—positioned in the bottom with respect to the area that is featured in the microscopy images. The scale bar represents 1 mm.

pulls the drain, a droplet of oleic acid with sodium oleate, towards the source, which means a net force is exerted on the drain droplet during absorption. As we will demonstrate, the absorption of the filaments by the drain pulls the drain droplet back towards the source, and matching the rates of the processes involved results in dynamic self-positioning of the source and drain droplets.

**Controlling repulsive forces driven by Marangoni flows.** We developed a kinetic model to assess how the Marangoni flow relies on both the rates of amphiphile release from the source droplet to the air–water interface ($\Phi_{source}$, in mol cm$^{-2}$ s$^{-1}$) as well as the rates of depletion from the air–water interface towards the drain droplet ($\Phi_{drain}$) and towards the underlying aqueous phase ($\Phi_{water}$). In the model (schematically presented in Fig. 2a and elaborated in the "Methods" section), $A_s$ equals the density of amphiphiles present in the source droplet and filaments, $\Gamma$ the density of amphiphiles present as surfactant at the air–water interface, $\theta$ the density of vacant sites at the air–water interface, $A_d$ the density of amphiphiles in the drain, and $A_m$ the density of amphiphiles dissolved in the underlying aqueous phase, respectively—all densities (in mol cm$^{-2}$) are averaged over the whole area. The irreversible release of amphiphiles from the source to the air–water interface is described as $\Phi_{source}$, with rate constant $k_1$. The irreversible depletion at the drain is described as $\Phi_{drain}$, with rate constant $k_2$ and rate equation $\Phi_{drain} = k_2(\Gamma - \Gamma_0)$, here $\Gamma_0$ is defined as the minimum value to which the density of $C_{12}E_4OH$ at the air–water interface can be decreased upon depletion by the drain solution. The depletion from the air–water

interface to the underlying aqueous phase is described as $\Phi_{water}$, with forward and backward rate constants $k_3$ and $k_{-3}$, respectively. The surface tension value is derived from the amphiphile density at the air–water interface via the Frumkin adsorption isotherm[53].

Simulations provided us with deeper insight into how the different processes affect the self-organization. As shown in Fig. 2b, the surface tension is predicted to drop from 72 to 28 mN m$^{-1}$ when the amphiphile source droplet is applied at $t = 0$ s. A steady state is established when the rate of amphiphile release from the source equals the rate of depletion to the underlying aqueous phase, $\Phi_{source} = \Phi_{water}$. Deposition of a drain droplet at $t = 10$ min switches $k_2 = 0$ (no drain) to $k_2 > 0$ (drain effective). When the amphiphile depletion at the drain is larger than the depletion to the underlying aqueous phase, a new steady-state sets in at a higher surface tension value, such that $\Phi_{source} \approx \Phi_{drain} > \Phi_{water}$ (Supplementary Fig. 3). In this drain-dominated depletion regime 1, a strong Marangoni flow emerges towards the drain (Fig. 2c), as experimentally observed when oleic acid was deposited as a drain: the surface tension increased, the drain instantly attracted filaments and pulled the droplets together, leading to merging of source and drain within a few seconds (Fig. 2e, Supplementary Movie 3, and Supplementary Fig. 4).

These observations suggest that a stable self-organization of source and drain droplets requires a reduced uptake of amphiphiles at the drain. When the depletion of the amphiphile is dominated by the depletion towards the underlying aqueous phase, rather than the drain, the Marangoni flow does not induce instantaneous merging of the source and drain droplets. To reduce the depletion rate (i.e., $k_2$) of the $C_{12}E_4OH$ amphiphile, we

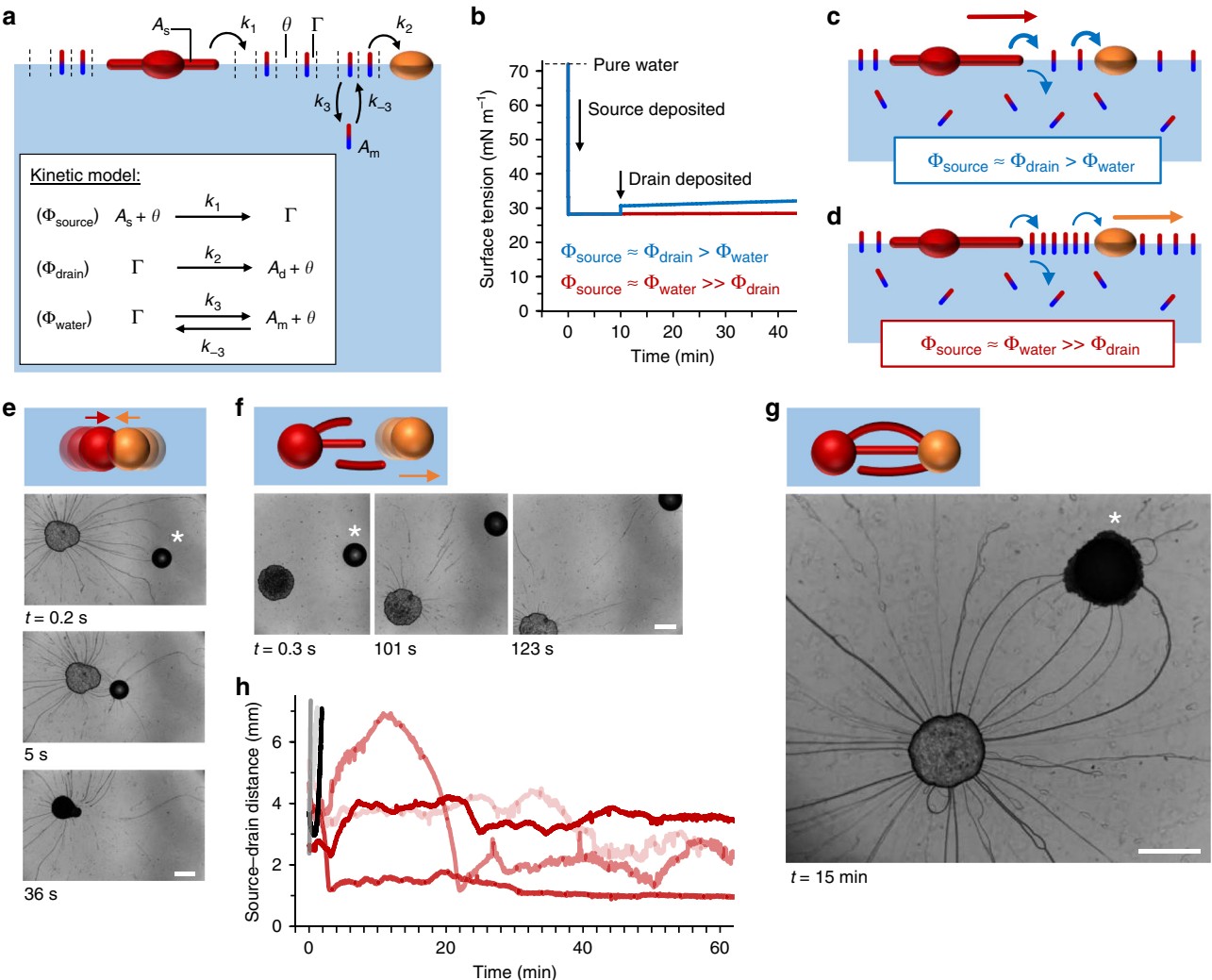

**Fig. 2 Droplet positioning relies on balanced rates of amphiphile release, depletion, and filament absorption. a** Kinetic model, describing $\Phi_{source}$, the release of amphiphiles from the source droplet (red sphere) to the air–water interface; $\Phi_{drain}$, the depletion of amphiphiles from the air–water interface to the drain droplet (orange sphere), and $\Phi_{water}$, the depletion of amphiphiles from the air–water interface to the underlying aqueous phase (blue). **b** Simulations of surface tension kinetics upon deposition of the source and drain droplets. The surface tension increases only if the uptake of amphiphiles at the drain is faster than the depletion towards the underlying aqueous phase (blue curve). **c** If $\Phi_{source} \approx \Phi_{drain} > \Phi_{water}$, the Marangoni flow is dominated by depletion at the drain. **d** If $\Phi_{source} \approx \Phi_{water} \gg \Phi_{drain}$, the Marangoni flow is dominated by depletion towards the underlying aqueous phase. **e** In the drain-dominated regime 1, a strong Marangoni flow emerged from the source droplet towards the drain (deposited at $t = 0$ s, indicated with an asterisk), and the drain merged with the source droplet. **f** In regime 2, the Marangoni flow was dominated by depletion towards the underlying aqueous phase and pushed the drain (deposited at $t = 0$ s) away from the source (deposited at $t = -34.5$ s). **g** In regime 3, the filaments were absorbed by the drain droplet, such that the drain was kept in position—despite the repulsive forces of the Marangoni flow. **h** Time-dependent center-to-center source–drain distance: the black and gray curves correspond to the experiments shown in **f** and Supplementary Fig. 5; the red curves to the experiments shown in **g** and Supplementary Fig. 6. The scale bars represent 1 mm.

included a sodium oleate surfactant (10 wt%) in the oleic acid drain solution. Upon deposition, the surface tension was observed to remain constant at 28 mN m$^{-1}$; below the minimum surface tension of sodium oleate at the air–water interface (30 mN m$^{-1}$; Supplementary Fig. 4). This observation indicates that the drain only provided a minor addition to the original depletion of $C_{12}E_4OH$ from the air–water interface, and $\Phi_{source} \approx \Phi_{water} \gg \Phi_{drain}$ (Fig. 2d and Supplementary Fig. 3). As a result, the Marangoni flow—dominated by the amphiphile release from the source and depletion towards the underlying aqueous phase—was observed to push the drain away from the source, following the aqueous phase dominated depletion regime 2 in Fig. 2f (Supplementary Movie 3 and Supplementary Fig. 5). A comparable suppression of the depletion rate was observed when

$C_{12}E_4OH$ (9 v/v%) was included in the oleic acid drain solution (Supplementary Fig. 4).

**Filaments attract the drain droplet upon absorption.** In order to establish a sustained self-organization where the droplets are kept in position, the repulsive nature of the Marangoni flow that pushes—when depletion is dominated by the aqueous phase—the drain away from the source, needs to be complemented by an attractive force. When the drain was deposited such that it hit a number of filaments originating from the source, these filaments were absorbed by the drain, and also new filaments were slowly drawn towards the drain—indicative of a small, but sustained Marangoni flow towards the drain (regime 3). Upon absorption

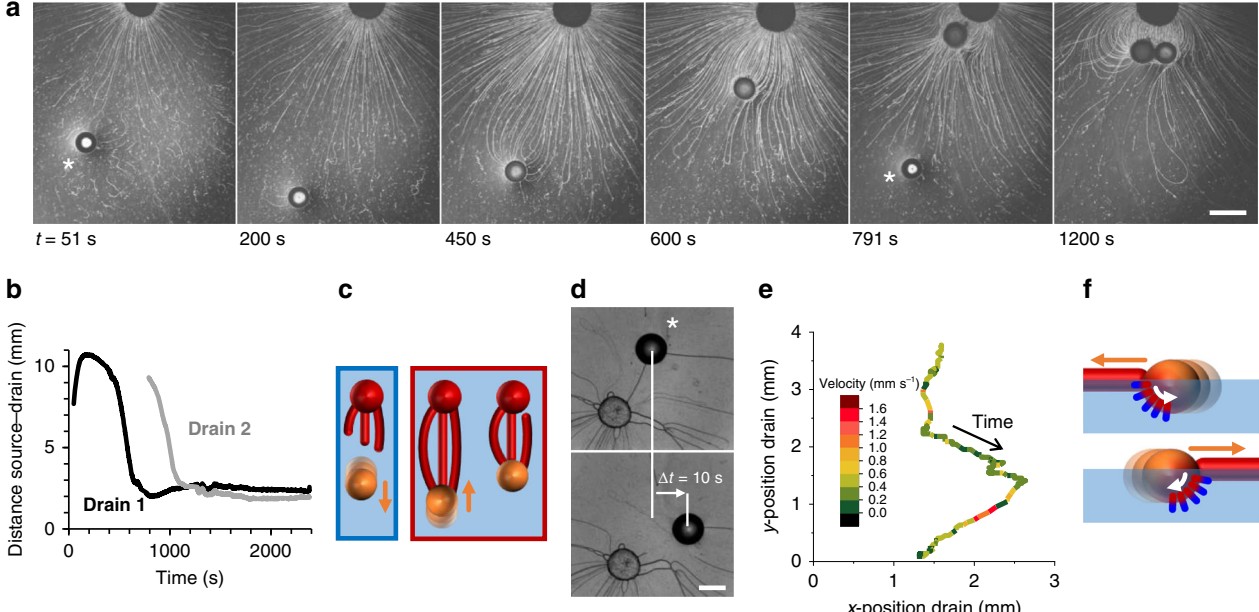

**Fig. 3 Filament absorption generates attractive forces at the drain droplet. a** Optical microscopy recording after subsequent deposition of source droplet ($t = 0$ s) and two 10 wt% sodium oleate in oleic acid drain droplets, indicated with asterisks, deposited at $t = 51$ and 791 s. **b** Distance source–drain vs. time, for both drain droplets in **a**. **c** Schematic representation of the self-organization, emerging from a balance between the Marangoni flow that repels the drain droplet (blue box), and filament absorption that generates attractive forces (red box). **d** Optical microscopy recording of a source and oleic acid drain droplet (asterisk) where the Marangoni flow from the source to the drain was suppressed by the inclusion of $C_{12}E_4OH$ in the aqueous phase. Upon absorption of a filament, the drain moved abruptly, over the course of $\Delta t = 10$ s, into the direction where the absorption took place, as shown by the $x$- and $y$-position (with respect to the source) of the drain droplet in **e**. **f** Upon absorption, the filament provides a non-uniform distribution of surfactants at the oil–water interface, driving an internal Marangoni flow that propels the droplet towards the absorption site. The scale bars represent 2 mm (**a**) and 1 mm (**d**), respectively.

of the filaments, the drain was kept in position and the droplets maintained a center-to-center separation of 1–4 mm over the course of >1 h, while filaments were slowly transferred from source to drain (Fig. 2g, h, Supplementary Fig. 6, and Supplementary Movie 3).

To corroborate the role of the myelin filaments in the attractive forces that pull the drain towards the source—opposed to the repulsive nature of the Marangoni flow—we followed a drain that was deposited further away from the source droplet (~7 mm), such that the filaments did not hit the drain immediately. As shown in Fig. 3a and Supplementary Movie 4, initially, the Marangoni flow pushed the drain away from the source droplet. However, filaments that arrived at the drain, following the Marangoni flow, were absorbed, and subsequently, the drain was pulled back towards the source, such that it stayed at a distance of ~2 mm from the source (Fig. 3b, c). A second drain droplet that was deposited subsequently was also drawn towards the source upon absorption of the filaments.

To assess the attractive forces exerted by the filament absorption events, we suppressed the Marangoni flow between the source and drain droplets upon including 4.3 mM of $C_{12}E_4OH$ in the aqueous solution, well beyond the critical micelle concentration of 0.1 mM[53]. The oleic acid drain attracts, upon depleting $C_{12}E_4OH$ from the air–water interface, filaments from all directions (Supplementary Movie 5). Importantly, absorption of filaments draws the drain shortly, over the course of the absorption event, into the direction where the absorption takes place (Fig. 3d, e). In the literature, it has been shown how an imbalance in the oil–water surface tension between the front and the rear of the droplet generates an internal Marangoni flow that propels the droplet towards the surfactant absorption site[35,36,43,54]. Absorption of a filament delivers an excess of

surfactant at the drain, and the transient non-uniform distribution of surfactants at the oil–water interface might drive the abrupt motion of the drain, as schematically shown in Fig. 3f.

Together, these results demonstrate that if the Marangoni flow is dominated by amphiphile depletion towards the underlying aqueous phase, and the filaments are absorbed by the drain, the positioning of free-floating source and drain droplets is maintained by the balance between (i) the Marangoni flow, which pushes the drain away, and (ii) the absorption of filaments, which causes the drain to be pulled toward the source.

**Autonomous positioning of multi-droplet systems**. To demonstrate how the well-balanced coupling between the Marangoni flow and the filament absorption at the drain drives autonomous positioning, we established the positioning of a free-floating source droplet in a ring of fixed drain droplets. The drain droplets were positioned at the tips of six pins, hexagonally positioned on a 5.8 mm diameter ring placed in the aqueous solution. When the source droplet was deposited, filaments started to grow and tethering of filaments to the drains enabled the positioning of the source droplet, which maintained its position within the hexagon over the course of >1 h (Fig. 4a–d, Supplementary Movie 6, and Supplementary Fig. 7). When no drain droplets were deposited at the pins, the source droplet moved around randomly within or even outside of the hexagon (Fig. 4e, f and Supplementary Fig. 8). Together, these results imply that the Marangoni flow drags the filaments towards the drains. Potentially, this flow also drags the source altogether towards one of the drains (which are fixed in position), and merging of the droplets has been observed when drain and source droplets contacted each other before the filament growth was started (Supplementary Fig. 9). However, when the source droplet is positioned in the

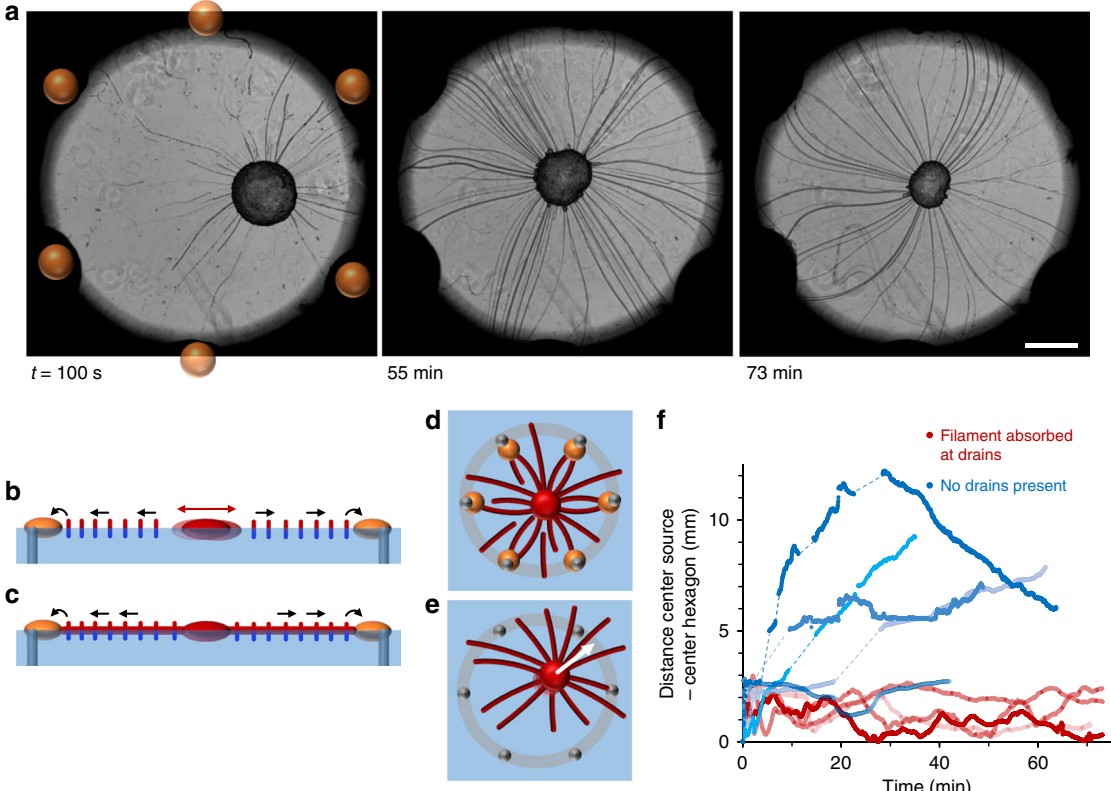

**Fig. 4 Autonomous positioning of dynamic assemblies in a ring of drain droplets. a** Optical microscopy recording of a free-floating source droplet self-positioning in between six drain droplets that were placed at the tips of hexagonally positioned pins—as indicated by the orange spheres at $t = 100$ s. **b**, **c** Schematic representation of the process: The Marangoni flow towards the drains moves the source droplet around (**b**), and drags the filaments towards the drains (**c**). (**d**) Upon tethering to the drains, the filaments stabilize the positioning of the source droplet. **e** In the absence of drain droplets, the source droplet moves around randomly within or outside the hexagon. **f** Time-dependent distance between the source droplet center and the hexagon center. The red curves correspond to the experiments shown in **a** and Supplementary Fig. 7, conducted under similar conditions; the blue curves correspond to experiments conducted without drain droplets (Supplementary Fig. 8). The scale bar represents 1 mm.

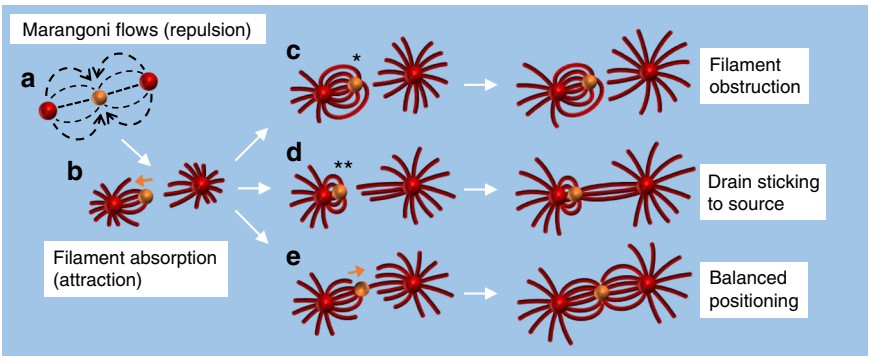

**Fig. 5 Proposed mechanism that mediates the positioning of interconnected droplet-filament systems. a** The Marangoni flows repel the drain (orange sphere) from the two sources (red spheres). **b** The Marangoni flow draws the filaments to the drain. Upon absorption of these filaments, the drain is drawn toward the source where these filaments originate from. **c** The filaments that are tethered to the drain obstruct the approach of filaments from the second source droplet (indicated with *). **d** The drain sticks to the source via capillary forces (indicated by **) that inhibit the drain to be drawn away by filaments arriving from the second source droplet. **e** Filaments that originate from the second source get absorbed at the backside of the drain, thereby generating attractive forces that draw the drain droplet back to the right, such that the Marangoni flows and the absorption of filaments from both sources together mediate positioning of the drain in between both sources.

middle of the hexagon, and filament connections have been established to all drains, the drag of the source favouring one drain is inhibited, as it requires breakage or disconnection of the filaments that are tethered to the opposite drains.

Finally, we studied how Marangoni flow and filament self-assembly coordinate complex positioning in situations where all elements are mobile, that is, free-floating. To this end, we deposited two source droplets of pure $C_{12}E_4OH$ and one drain droplet at the air–water interface. One would expect that the Marangoni flows emerging from both source droplets repel the drain (Fig. 5a). Subsequently, as schematically shown in Fig. 5b, once the filaments arrive at the drain, attractive forces are

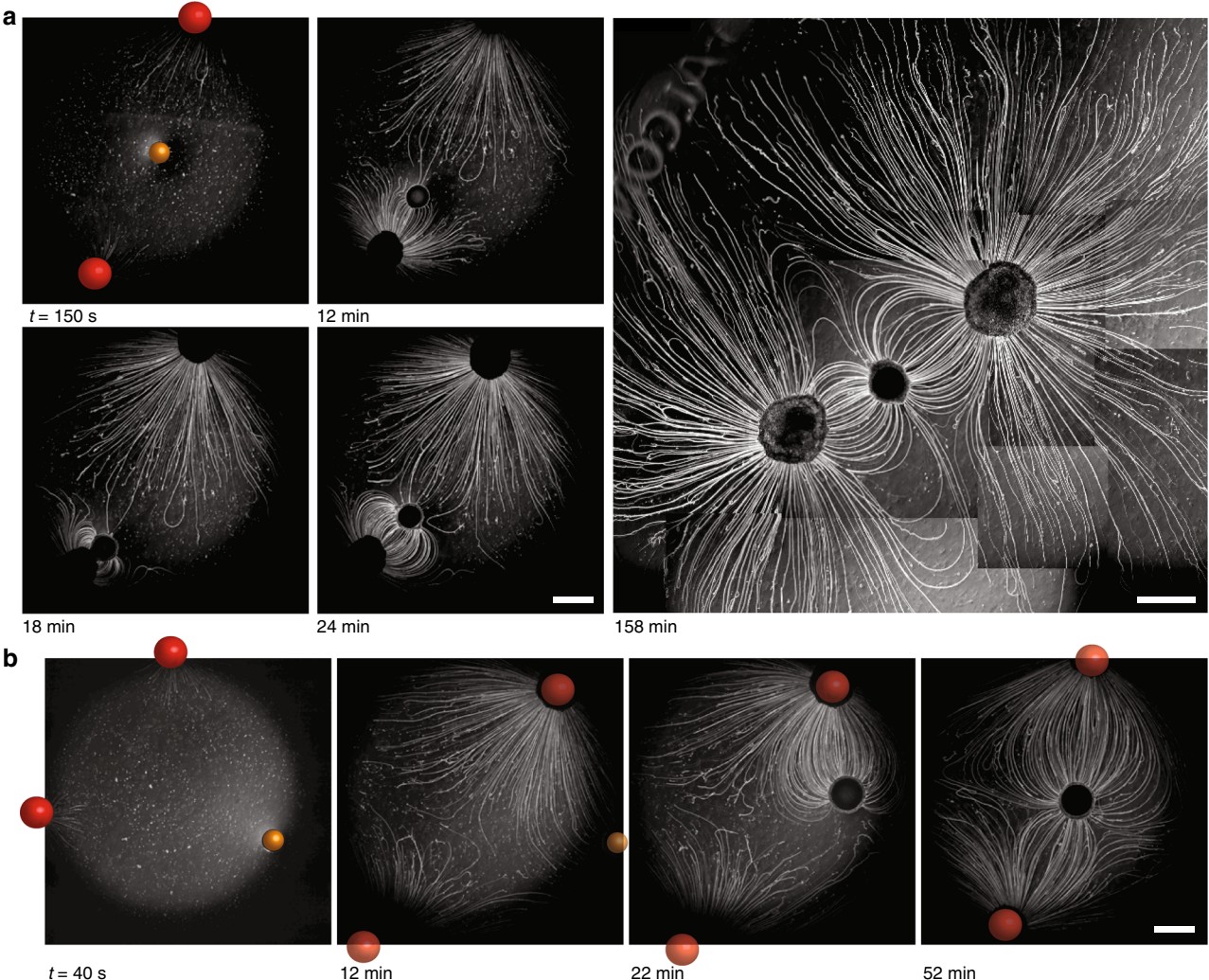

**Fig. 6 Capture and positioning of droplets by balancing Marangoni flow and absorption of filaments. a** Optical microscopy recording of the positioning process of a drain droplet (orange sphere) in between two source droplets (red spheres) after deposition at the air–water interface. As the filaments grew from the source droplets and tethered the drain droplet, they established a stable, interconnected droplet network. The wide-field image acquired at $t =$ 158 min is composed of multiple microscopy images. **b** Optical microscopy recording of the positioning process after deposition of the drain droplet off-centered to the two source droplets. Absorption of filaments drew the drain towards the top source, and subsequently, upon absorption of filaments from the bottom source, the drain was positioned in between the two source droplets. The scale bars represent 2 mm.

generated upon absorption, which draw the drain towards the respective source droplet—in analogy to the results shown in Fig. 3a. Next, we hypothesized that if the drain droplet, while being tethered via the filaments to the first source, can be approached by filaments from the second source droplet, absorption of these filaments draws the drain back towards the second source. If the approach of filaments from the second source is not obstructed by the filaments from the first source droplet (Fig. 5c), and the retraction of the drain droplet from the first source droplet is not impaired by strong capillary forces that stick the drain droplet to the source (Fig. 5d), a well-balanced linear self-organization of source and drain droplets is obtained, as schematically shown in Fig. 5e. We observed, while allowing the self-organization to take place under similar conditions in 12 experiments, the linear self-organization as shown in Fig. 6a and Supplementary Movie 7 for $n = 5$ times (Supplementary Figs. 10 and 11), as well as behavior schematically shown in Fig. 5c for $n = 4$ times (Supplementary Fig. 12) and behavior schematically shown in Fig. 5d for $n = 3$ times (Supplementary Fig. 13). The linear self-organization has been observed to remain stable over

>5 h (Supplementary Fig. 11). NB the increased filament density was obtained by increasing the $C_{12}E_4OH$ content to 100% in the source droplets.

Importantly, the positioning relies on the balance between repulsion and attraction. When larger amounts of sodium oleate were included in the oleic acid drain (15 wt%, 20 wt%; Supplementary Fig. 14), a smaller number of filaments tethered to the drain, indicating that sodium oleate suppresses the capacity of the drain to attract and absorb filaments. This suppression also enabled to position two drain droplets in between two source droplets, a conformation that was not established with 10 wt% sodium oleate (Supplementary Fig. 15). Finally, to assess the versatility of this positioning mechanism, we deposited a drain droplet off-centered to the source droplets. As shown in Fig. 6b, Supplementary Movie 8, and Supplementary Figs. 16 and 17 ($n = 7$), absorption of filaments from both sources resulted in attractive forces that drew the drain back such that it, ultimately, ended up in between both source droplets—in analogy to the mechanism outlined in Fig. 5a, b, e.

## Discussion

Concentration gradients, such as the surface tension gradients that drive the Marangoni flow involved in our system, offer a general design principle to provide spatial differentiation at mesoscopic length scales. We note that droplet-based systems—often in combination with dissipative surfactant systems, nanostructures or out-of-equilibrium assemblies—are gaining popularity as a platform to explore the emergence of complex, collective behavior in multi-component systems, with interest varying from origin-of-life contexts[55,56] to multiphase reactors[57], sensors[58,59], reconfigurable objects[60,61], optics[62], and maze solving[43,63]. The concept of self-assembling, wire-like structures that are transferred among droplets and concomitantly aid in their positioning can become useful in scenarios where transfer of signals adds programmable coordination in self-organization behavior. Additional feedback mechanisms, for example, by delivering chemicals that enhance or counteract a local depletion of surfactants, might enable autonomously operating systems that form and re-wire connections along 2D substrates—providing potential in dynamic, smart materials.

In summary, we have demonstrated a system that displays mesoscale self-organization, driven by self-sustained out-of-equilibrium fluxes of amphiphiles that mediate the positioning of their origin. Marangoni flows push amphiphile source and drain droplets apart when the rates of amphiphile release and depletion from the air–water interface are in the right regime, such that the depletion is dominated by migration toward the underlying aqueous phase. At the same time, the amphiphile molecules self-assemble into filaments that originate from buckling of amphiphile bilayers and get extruded from the floating amphiphile source droplets by the Marangoni flow. As the filaments get absorbed by the drain, the drain droplet is attracted back towards the source. Together, the repulsive forces from the Marangoni flow and the attractive forces from the filament absorption keep the droplets in position, such that the positioning of interconnected filament-droplet networks is sustained.

## Methods

**Materials**. Tetra(ethylene glycol) monododecyl ether (≥98.0%) was purchased from Sigma-Aldrich and Santa Cruz Biotechnology Inc. (Dallas, TX). Sodium alginate and (+)-sodium L-ascorbate (≥99.0%) were purchased from Sigma-Aldrich, sodium chloride (99.6%) from Fisher Chemical, oleic acid from Fisher Chemical (≥70.0%) and Fluorochem (95%), ethanol (absolute for analysis) and sodium benzoate from Merck, and sodium oleate (95%) from ABCR GmbH (Karlsruhe, Germany). All materials were used as received.

**Source and drain solutions**. The $C_{12}E_4OH$ source solution was prepared by mixing 40 v/v% tetra(ethylene glycol) monododecyl ether $C_{12}E_4OH$, 15 v/v% ethanol, and 45 v/v% water (unless stated otherwise). The drain solutions of oleic acid with sodium oleate were prepared by dissolving sodium oleate in oleic acid upon sonication. The sodium alginate solution was prepared by dissolving sodium alginate in water (6.25 mg mL$^{-1}$). For the experiments that involved a drain droplet, also (+)-sodium L-ascorbate (17 mM) was included in the sodium alginate solution, as the inclusion of a sodium salt was observed to enhance the stability of the drain droplet (Supplementary Fig. 18). The viscosity of the (+)-sodium L-ascorbate/sodium alginate solution was determined at ambient temperature using a TA instruments Discovery Hybrid Rheometer-2 (5.8 mPa.s), and the pH of the solution was measured to be 6.5.

**Study self-organization with microscopy**. Optical microscopy images and movies were acquired with an Olympus IX71 dark-field inverted microscope equipped with a Phantom Vision camera, and an Olympus IX73 dark-field inverted microscope equipped with a Point Grey Grasshopper3 camera. A ×2 objective was used, unless stated otherwise. The frame rate was chosen appropriately; movies over the time course of ~1 h were acquired at a frame rate of 1 fps. The composed microscopy images in Fig. 6a and Supplementary Fig. 15 were assembled manually. The time-dependent positioning traces of the free-floating source and drain droplets were analyzed using Matlab (R2017a, imfindcircles function) and the TrackMate ImageJ algorithm[64].

To initiate the growth of the filaments, the amphiphile solution droplets were deposited with a Gilson pipette at the interface of the sodium alginate solution in a polystyrene petri dish (lid of a Falcon 35 mm dish, diameter 38 mm, height 4.5 mm, used as received). In order to prevent the droplet from moving towards the solution meniscus at the edge of the petri dish, the petri dish was filled completely with sodium alginate solution, such that a convex air–water interface was formed (5.5 mL, solution height 4.85 mm). Prior to the deposition of the amphiphile droplet, the surface tension of the sodium alginate solution was decreased to ~28 mN m$^{-1}$ by adding a trace amount of amphiphile solution to the interface: this avoids the source droplet to be exposed to large surface tension gradients that rapidly tear apart the droplet upon deposition. All experiments were performed at room temperature, as a minimum temperature of ~20 °C has been reported in the literature for the laminar phase of $C_{12}E_4OH$, which is required to form the filaments[44]. The formation of the laminar phase of $C_{12}E_4OH$ in our experiments was verified by the opaque appearance, to the naked eye, of the $C_{12}E_4OH$ droplet after deposition at the sodium alginate solution.

In the experiment shown in Fig. 1e, 0.5 µL of the amphiphile solution was deposited. The images shown in Fig. 1d were acquired at a larger magnification (×10 objective); to keep the amphiphile droplet in position while acquiring those images, the droplet (1.0 µL) was positioned at a metal pin (folded paperclip, diameter 0.75 mm) that was inserted in the sodium alginate solution. In the experiment shown in Fig. 1f, 0.5 µL of the amphiphile solution as well as 0.5 µL of the drain solution (10 wt% sodium oleate in oleic acid) were deposited with a Gilson pipette at the sodium alginate solution.

In the experiment shown in Fig. 2e, 0.5 µL of the amphiphile solution was deposited, and 0.5 µL oleic acid. In Fig. 2f, g, 0.5 µL of the amphiphile solution was deposited, and 0.5 µL of the drain solution (10 wt% sodium oleate in oleic acid).

In the experiment shown in Fig. 3a, 1.0 µL $C_{12}E_4OH$ was deposited, followed by two subsequent droplets of drain solution (10 wt% sodium oleate in oleic acid, 1.0 µL). The medium consisted of 5.5 mL 6.25 mg mL$^{-1}$ sodium alginate and 17 mM sodium chloride in MilliQ. The optical microscopy images were acquired at ×1.25 magnification. In the experiment shown in Fig. 3d, a $C_{12}E_4OH$ source droplet (1.0 µL) and an oleic acid drain droplet (1.0 µL) were deposited subsequently at an aqueous sodium alginate solution (6.25 mg mL$^{-1}$) that included $C_{12}E_4OH$ (4.3 mM) to suppress the Marangoni flow from the source droplet.

In Fig. 4, a 3D printed hexagon (see Supplementary Fig. 7, 3D printed via fused filament technique using polylactic acid, inner ring diameter 5.8 mm, pin height 5.2 mm, pin thickness 1.8 mm) was placed in the petri dish. The sodium alginate solution was applied such that the hexagon pins were touching the air–water interface and the air–water interface formed a concave meniscus at the hexagon pins. Then, the surface tension of the sodium alginate solution was decreased by touching the interface with a pipette tip with amphiphile solution; the drain droplets (1.0 µL, 10 wt% sodium oleate in oleic acid) were deposited at the tips of the pins, and the amphiphile solution (1.0 µL) was deposited in the center of the hexagon.

In the experiments shown in Fig. 6, first two 1.0 µL droplets of pure $C_{12}E_4OH$ were deposited simultaneously at the sodium alginate solution, using a Gilson 8-channel multipipette. After 25 s, 1.0 µL of the drain solution (10 wt% sodium oleate in oleic acid) was deposited at approximately equal distance to both source droplets (in the center of both source droplets for Fig. 6a; off-centered to both source droplets in Fig. 6b). The optical microscopy images were acquired at ×1.25 magnification. The experiments shown in Supplementary Figs. 10–17 were conducted under similar conditions.

**Surface tension experiments**. The surface tension kinetics (Supplementary Figs. 1, 2, and 4) were acquired using a KSV Instruments LTD surface tensiometer that measures the force exerted by the air–water meniscus on a platinum Wilhelmy plate (19.62 mm×10 mm) inserted into the aqueous solution. For the experiment shown in Supplementary Fig. 1 that was conducted with an air–water interface area $a$ of 244 cm$^2$, first a Teflon Langmuir–Blodgett (LB) trough (7.5 cm × 32.5 cm) was cleaned upon subsequent rinsing with water, ethanol and water, and drying using pressurized nitrogen. Next, the trough was filled with MQ water, and the barriers of the LB setup were closed slowly in order to compress a monolayer of potential contaminants occupying the air–water interface. Next, the upper layer of the water in between the barriers was removed upon suction, new water was added from behind the barriers to substitute the removed water, and the barriers were opened again. The Wilhelmy plate was cleaned with water and ethanol, flamed until red hot, and inserted into the water, such that the water wetted the plate and formed a meniscus. The surface tension was set at 72 mN m$^{-1}$ for pure water, and after 2 min, the amphiphile solution was deposited at the air–water interface, ~10 cm away from the Wilhelmy plate. For the experiment performed with $a = 59.4$ cm$^2$, a polystyrene petri dish (Falcon 87 mm × 18 mm, used as received) was filled with 80 mL MQ water (solution height 1 cm), and 1.0 µL of the amphiphile solution was deposited. For the experiment performed with $a = 9.6$ cm$^2$, a polystyrene petri dish (Falcon 35 mm, height 9.5 mm, used as received) was filled with 5 mL MQ water (solution height 5.2 mm) and 0.5 µL of the amphiphile solution was deposited.

For the experiment shown in Supplementary Fig. 2, where the surface tension kinetics were followed upon depositing an amphiphile source droplet at a sodium alginate solution (6.25 mg mL$^{-1}$), we used an LB trough ($a = 244$ cm$^2$), following the procedure described above.

For the experiments shown in Supplementary Fig. 4, where the surface tension kinetics were followed upon successively depositing an amphiphile source droplet

and a drain droplet, polystyrene petri dishes (Falcon, diameter 35 mm, height 9.5 mm, used as received, $a = 9.6$ cm$^2$) were filled with 5 mL MQ water (solution height 5.2 mm). The source and drain droplets were deposited such that they moved towards the meniscus at the edge of the petri dish, and did not get into contact with each other.

**Kinetic model to simulate surface tension kinetics**. The kinetic model that predicts the surface tension during the self-organization process involves the irreversible release of the amphiphile from the source droplet to the air–water interface, described with rate constant $k_1$ (Fig. 2a, $\Phi_{\text{source}}$); the irreversible depletion of amphiphile at the drain droplet, described with rate constant $k_2$ ($\Phi_{\text{drain}}$), and the reversible depletion of amphiphile from the air–water interface to the bulk phase of the aqueous solution, described with forward and backward rate constants $k_3$ and $k_{-3}$, respectively ($\Phi_{\text{water}}$). We simulated the density of amphiphile present as individual surfactant at the air–water interface ($\Gamma$), based on which we computed the surface tension value (vide infra). Furthermore, $A_s$ equals the density of amphiphiles present in both the source droplet as well as the filaments, $\theta$ the density of vacant sites at the interface, and $A_m$ the density of amphiphiles in the bulk phase of the aqueous solution. All density parameters are in mol cm$^{-2}$, and averaged over the whole area: our model does not involve spatial specification.

Based on the reaction equations presented in Fig. 2a, we defined the following rate equations:

$$\frac{dA_s}{dt} = -k_1 A_s \theta \quad \text{amphiphile in source droplet and filaments,} \tag{1}$$

$$\frac{d\theta}{dt} = -k_1 A_s \theta + k_3 \Gamma - k_{-3} A_m \theta + k_2 (\Gamma - \Gamma_0) \quad \text{vacant sites at the interface,} \tag{2}$$

$$\frac{d\Gamma}{dt} = k_1 A_s \theta - k_3 \Gamma + k_{-3} A_m \theta - k_2 (\Gamma - \Gamma_0) \quad \text{amphiphile at air} - \text{water interface,} \tag{3}$$

$$\frac{dA_m}{dt} = k_3 \Gamma - k_{-3} A_m \theta \quad \text{amphiphile in the underlying aqueous phase.} \tag{4}$$

Here, $\Gamma_0$ is defined as the minimum value to which the density of $C_{12}E_4OH$ at the air–water interface can be decreased upon depletion by the drain solution.

The overall change in the amphiphile density at the air–water interface, $\Gamma$, can be described via:

$$\frac{d\Gamma}{dt} = \Phi_{\text{source}} - \Phi_{\text{drain}} - \Phi_{\text{water}}, \tag{5}$$

where $\Phi_{\text{source}}$ equals the rate of amphiphile release from the source, $\Phi_{\text{water}}$ the rate of amphiphile depletion to the underlying aqueous phase, and $\Phi_{\text{drain}}$ the rate of amphiphile depletion to the drain, in mol cm$^{-2}$ s$^{-1}$:

$$\Phi_{\text{source}} = k_1 A_s \theta, \tag{6}$$

$$\Phi_{\text{drain}} = k_2 (\Gamma - \Gamma_0), \tag{7}$$

$$\Phi_{\text{water}} = k_3 \Gamma - k_{-3} A_m \theta. \tag{8}$$

To compute the surface tension $\gamma$ (in mN m$^{-1}$) based on $\Gamma$, we used the Frumkin isotherm, as reported by Hsu et al.[53]:

$$\gamma = \gamma_0 + 10^7 \times \Gamma_\infty RT \left( \ln\left(1 - \frac{\Gamma}{\Gamma_\infty}\right) - \frac{K}{2}\left(\frac{\Gamma}{\Gamma_\infty}\right)^2 \right) \quad \text{Frumkin isotherm.} \tag{9}$$

Here, $\gamma_0$ represents the surface tension of the clean air–water interface (72 mN m$^{-1}$ for pure water), $R$ the gas constant, $T$ the temperature ($T = 293$ K), $K$ the adsorption cooperativity factor ($K = 1.875$), and $\Gamma_\infty$ the maximum surface concentration ($\Gamma_\infty = 4.663 \times 10^{-10}$ mol cm$^{-2}$)[53].

To simulate the surface tension kinetics, we defined the starting values for $A_s$, $\theta$, $\Gamma$, and $A_m$. Initially, all amphiphile molecules are present in the source droplet, implying that $A_s(t = 0)$ can be derived from the volume of the amphiphile source droplet and the area of the air–water interface; $\Gamma(t = 0) = 0$ and $A_m(t = 0) = 0$. The density of vacant positions available at the air–water interface was derived from the literature data on the equilibrium density of amphiphiles present at the air–water interface just beyond the critical micelle concentration. Based on the Frumkin isotherm, and data reported by Hsu et al.[53], we derived $\theta(t = 0) = 4.44 \times 10^{-10}$ mol cm$^{-2}$.

The simulations shown in Fig. 2b and Supplementary Fig. 3 were performed by solving the system of differential Eqs. (1–4) using Matlab (R2017a, ode15s solver), with $k_1 = 9.6 \times 10^7$ cm$^2$ mol$^{-1}$ s$^{-1}$; $k_3 = 1.2 \times 10^{-2}$ s$^{-1}$; $k_{-3} = 1.09 \times 10^7$ cm$^2$ mol$^{-1}$ s$^{-1}$; $\Gamma_0 = 4.33 \times 10^{-10}$ mol cm$^{-2}$ and $A_s(t = 0) = 5.42 \times 10^{-8}$ mol cm$^{-2}$. For the regime where $\Phi_{\text{source}} \approx \Phi_{\text{drain}} > \Phi_{\text{water}}$, the simulation was performed with $k_2 = 5$ s$^{-1}$; for the regime where $\Phi_{\text{source}} \approx \Phi_{\text{water}} \gg \Phi_{\text{drain}}$, the simulation was performed with $k_2 = 0.05$ s$^{-1}$.

## Data availability
The data that support the findings of this study presented within the article and its Supplementary information files are available from the corresponding author upon reasonable request.

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

## Acknowledgements

We thank the Dutch Research Council (NWO, Veni grant no. 722.016.009 and START-UP grant no. 740.018.003) as well as the Dutch Ministry of Education, Culture and Science (Gravitation program 024.001.035) for financial support. We also thank J. Aizenberg for stimulating discussions.

## Author contributions

P.A.K. conceived and supervised the project, W.T.S.H. contributed to the multi-droplet self-positioning concept, which was experimentally established by A.v.d.W. and M.W. M.W., A.v.d.W., S.M.C.S., and P.A.K. performed the experiments. P.A.K. developed the model. P.A.K., W.T.S.H., and M.W. wrote the paper.

## Competing interests

The authors declare no competing interests.
