## [Peer Review File · Nature Communications]

REVIEWERS' COMMENTS:

Reviewer #1, comments to the author

I previously reviewed the work of Arno van der Weijden et Al. for the journal "Nature Nanotechnology" and I raised several issues that, in my opinion, prevented the publication of the manuscript.

The authors made a great effort in addressing my comments and their revisions resulted convincing.

I think the quality of the manuscript benefited also from the comments of the other reviewers, that the authors addressed as well.

Now the work is set in a proper framework and several details (use of the oleate, role of filaments, etc.) were clarified and added to the discussion. The novelty of the work with respect to the existing literature was also clearly stated.

I feel that Nature Communications is the proper journal where to publish this interesting work.

Therefore I support the publication of this manuscript in this revised form.

Reviewer #2, comments to the author

The authors have done a great job in clarifying the reviewers' questions. The manuscript is highly suitable for Nature Communications. I recommend publication without further changes.

Reviewer #3, comments to the author:

All reviewer comment have been addressed, and I would recommend publication of the manuscript in it's current form. I would also suggest to publish the reply to the reviewers, as it is quite rich and might serve as a background for readers interested in certain details.

Responses to Comments of Reviewers:

Reviewer #1, comments to the author

I previously reviewed the work of Arno van der Weijden et Al. for the journal "Nature Nanotechnology" and I raised several issues that, in my opinion, prevented the publication of the manuscript. The authors made a great effort in addressing my comments and their revisions resulted convincing.

I think the quality of the manuscript benefited also from the comments of the other reviewers, that the authors addressed as well. Now the work is set in a proper framework and several details (use of the oleate, role of filaments, etc.) were clarified and added to the discussion. The novelty of the work with respect to the existing literature was also clearly stated.

I feel that Nature Communications is the proper journal where to publish this interesting work.

Therefore I support the publication of this manuscript in this revised form.

We thank the reviewer for these positive comments on our work, and for earlier suggestions that really helped us to improve our manuscript.

Reviewer #2, comments to the author

The authors have done a great job in clarifying the reviewers' questions. The manuscript is highly suitable for Nature Communications. I recommend publication without further changes.

We thank the reviewer for these positive statements.

Reviewer #3, comments to the author:

All reviewer comment have been addressed, and I would recommend publication of the manuscript in it's current form. I would also suggest to publish the reply to the reviewers, as it is quite rich and might serve as a background for readers interested in certain details.

We thank the reviewer for these positive statements. We are pleased to read that the reviewer appreciates our previous reply to the reviewers. In fact, we are currently preparing more detailed follow-up publications on our system.